# In Vitro Toxicity, Antioxidant, Anti-Inflammatory, and Antidiabetic Potential of *Sphaerostephanos unitus* (L.) Holttum

**DOI:** 10.3390/antibiotics9060333

**Published:** 2020-06-18

**Authors:** Marimuthu Alias Antonisamy Johnson, C. Xavier Madona, Ray S. Almeida, Natália Martins, Henrique D. M. Coutinho

**Affiliations:** 1Centre for Plant Biotechnology, Department of Botany, St. Xavier’s College (Autonomous), Palayamkottai, Tamil Nadu 627 002, India; madonaxavier@gmail.com; 2Laboratory of Microbiology and Molecular Biology (LMBM), Regional University of Cariri-URCA, Crato 63105-000, CE, Brazil; rayalmeidasilva2306@gmail.com (R.S.A.); hdmcoutinho@gmail.com (H.D.M.C.); 3Faculty of Medicine, University of Porto, Alameda Prof. Hernani Monteiro, 4200-319 Porto, Portugal; 4Institute for research and Innovation in Health (i3S), University of Porto, Rua Alfredo Allen, 4200-135 Porto, Portugal

**Keywords:** *Sphaerostephanos unitus*, secondary metabolites, biopotency, fern

## Abstract

Pteridophytes have been widely used in several systems of medicine. Several reports have increasingly assessed their bioactive effects, but for *Sphaerostephanos unitus* (L.) Holttum, only its antibacterial potential has been assessed. In this sense, the present study was carried out to reveal the phytochemical profile and to determine the toxicity, antioxidant, antidiabetic, and anti-inflammatory potential of *S. unitus*. Brine shrimp lethality, 2,2-diphenyl-1-picrylhydrazyl (DPPH) radical scavenging activity, phosphomolybdenum assay, superoxide radical scavenging activity, 2,2-azino-bis (3-ethylbenzothiazoline-6-sulfonic acid) assay (ABTS), and in vitro α-amylase inhibitory and membrane stabilization assays were applied. *S. unitus* extract toxicity showed variable mortality percentages, with LC_50_ values ranging from 4 to 30 mg/mL. DPPH radical scavenging effects of *S. unitus* extracts were as follows: methanol > acetone > petroleum ether > chloroform. *S. unitus* acetone extract displayed the strongest phosphomolybdenum reduction (10 ± 2 mg Ascorbic Acid Equivalent/g). The studied extracts also revealed efficient, superoxide scavenging effects in a dose-dependent manner. In *S. unitus*, the highest ABTS radical scavenging rate was observed in the chloroform extract (3000 ± 40 µmol/g). The *S. unitus* anti-inflammatory effect was as follows: petroleum ether > chloroform > methanol > acetone. In *S. unitus* extract, the highest percentage of α-amylase activity (80%) was observed for the petroleum ether extract (25 µg/mL). Faced with these findings, further studies should be performed to isolate and identify the *S. unitus* compounds responsible for their antioxidant, antidiabetic and anti-inflammatory effects.

## 1. Introduction

Lycophytes and ferns are a neglected group of plants in biodiversity as far as their economic value is concerned. The world flora of lycophytes and ferns harbor about 12,000 species, of which 1000 species from 70 families and 191 genera occur in India [1]. 

Pteridophytes are mostly distributed in the Himalayas and the Western Ghats. More than 300 species of ferns and fern allies are reported from the Western Ghats, South India [2]. Pteridophytes have been successfully used in different systems of medicines like Ayurvedic, Unani, and homeopathic, among others. Kaushik [3] emphasized the ethnobotanical importance of ferns of Rajasthan, India. Recently, Yumkham et al. [4] reported some edible pteridophytes as vegetables and medicines. However, despite its folk medicinal uses, its biological effects are far from being consolidated, from preclinical to clinical studies. Scientific data available have reported the antioxidant effects of pteridophytes [5,6,7,8,9,10,11,12,13,14]. On the other hand, the multitude of adverse side effects derived from the use of antidiabetic drugs has led to pteridologists and pharmacists to focus their research on producing alternative safer drugs, without any side effects. However, few reports are available on the antidiabetic potential of pteridophytes. Chai et al. [15] evaluated the antidiabetic potential of five selected edible and medicinal ferns, while Paul et al. [16], Basha et al. [17], and Tanzin et al. [18] reported the antidiabetic potential of *Adiantum philippense*, *Angiopteris evecta, Actinopteris radiata,* and *Christella dentata.* In addition, glancing at the most commonly used drugs to reduce inflammatory conditions, nonsteroidal anti-inflammatory drugs (NSAIDs) are on the top, not only for sales but also for adverse effects, especially since they can cause gastric irritation. Hence, in recent years, medicinal plants have become a subject of interest for drug development owing to their phytochemical constituents and therapeutic potential. Few reports are available on the anti-inflammatory potential of pteridophytes [19,20,21,22,23,24,25]. Paulraj et al. [26] reported the antibacterial ability of *Sphaerostephanos subtruncatus* and *S. unitus*, but no reports are available on the antioxidant, antidiabetic, anti-inflammatory, and cytotoxic effects of *Sphaerostephanos unitus* (L.) Holttum. In this sense, the present study aims to reveal the qualitative and quantitative secondary metabolites profile of *S. unitus* and to determine its toxicity, antioxidant, antidiabetic, and anti-inflammatory potential.

## 2. Results

### 2.1. Phytochemical Analysis

Among the various tested extracts, petroleum ether, acetone, chloroform, and methanol extracts of *S. unitus* showed the presence of all the studied metabolites (28/32) except alkaloids (Table 1). 

### 2.2. Metabolites Quantification

Total phenolic, tannins, steroids, flavonoids, and terpenoids obtained from the various extracts of *S. unitus* are are shown in Table 2. The total phenolic contents of *S. unitus* were as follows: methanol (300 ± 10 mg GAE/g DW) > chloroform (90 ± 20 mg GAE/g DW) > petroleum ether (60 ± 20 mg GAE/g DW) > acetone (50 ± 5 mg GAE/g DW). Among the various *S. unitus* extracts, the methanol extract showed the maximum amount of tannins (50 ± 2 mg RE/g DW), with the least amount from the chloroform extract (10 ± 1 mg RE/g DW). The maximum extractable steroids were observed in chloroform (0.2 ± 0.03 mg/g), followed by acetone (0.2 ± 0.01 mg/g) and petroleum ether (0.1 ± 0.04 mg/g). The methanol extract of *S. unitus* revealed the lowest level of steroids (0.1 ± 0.01 mg/g). With regards to total extractable flavonoids in *S. unitus*, it was as follows: methanol > petroleum ether > chloroform > acetone, while an inverse trend was found for extractable terpenoids: chloroform > acetone > petroleum ether > methanol.

### 2.3. Toxicity of S. unitus

A variable percentage of mortality was stated for *S. unitus*, with LC_50_ values ranging from 4 to 30 mg/mL being found the following order of toxicity: methanol > acetone > petroleum ether > chloroform. The LC_50_, LC_75_ and LC_90_ values of the *S. unitus* were calculated and are recorded in Figure 1.

### 2.4. Antioxidant Activity

#### 2.4.1. DPPH Radical Scavenging Activity

Free radical scavenging activity of the different *S. unitus* extracts are shown in Table 3. The DPPH radical scavenging activity of *S. unitus* extracts was as follows: methanol > acetone = petroleum ether = chloroform. The mean difference was considered statistically significant at the 0.05 level. The *t*-values for the extracts and DPPH radical scavenging activity of *S. unitus* were as follows: acetone (*t* = 0.000), chloroform (*t* = 0.002), methanol (*t* = 0.000), and petroleum ether (*t* = 0.012) extracts.

#### 2.4.2. Phosphomolybdenum Assay

The results of phosphomolybdenum assay for the different *S. unitus* extracts are illustrated in Table 4. *S. unitus* acetone extract displayed the strongest phosphomolybdenum reduction (10 ± 2 mg AA/g). The mean difference was considered statistically significant at the 0.05 level. The *t*-values for the extracts and phosphomolybdenum reduction of *S. unitus* were as follows: acetone (*t* = 0.049), chloroform (*t* = 0.004), methanol (*t* = 0.006), and petroleum ether (*t* = 0.021) extracts.

#### 2.4.3. Super Oxide Scavenging Activity

*S. unitus* extracts revealed efficient, superoxide scavenging effects in a dose-dependent manner (Table 5). In *S. unitus,* all tested extracts showed maximum inhibition percentages, ranging from 70 to 100%. The superoxide scavenging activity of *S. unitus* was as follows: methanol > chloroform > acetone > petroleum ether. The mean difference was considered statistically significant at the 0.05 level. The *t*-value for the extracts and superoxide scavenging activity of *S. unitus* were as follows: acetone (*t* = 0.001), chloroform (*t* = 0.000), methanol (*t* = 0.000), and petroleum ether (*t* = 0.000) extracts.

#### 2.4.4. ABTS Radical Scavenging Assay

In *S. unitus*, the highest ABTS radical scavenging rate was observed for the chloroform extract (3000 ± 40 µmol/g), followed by petroleum ether (2000 ± 70 µmol/g), methanol (2000 ± 200 µmol/g), and acetone (2000 ± 600 µmol/g) extracts (Table 6). The mean difference was considered statistically significant at the 0.05 level. The *t*-values for the extracts and ABTS radical scavenging activities of *S. unitus* were as follows: acetone (*t* = 0.033), chloroform (*t* = 0.049), methanol (*t* = 0.044), and petroleum ether (*t* = 0.029) extracts. Correlation values between the metabolites’ concentration and antioxidant activity of *S. unitus* was calculated and presented in Table 7.

### 2.5. Anti-inflammatory Activity

Stabilization of red blood cell (RBC) membranes was studied to further establish the mechanism of anti-inflammatory action of *S. unitus* extracts. The results are evidence of the membrane stabilization effects of *S. unitus* extracts as an additional mechanism for their anti-inflammatory effect (Figure 2). The studied extracts also inhibited heat-induced hemolysis of RBCs, with anti-inflammatory effects of *S. unitus* extracts being ranked in the following order: petroleum ether > chloroform > methanol > acetone (Figure 2). The mean difference was considered statistically significant at the 0.05 level. The *t*-values for the extracts and anti-inflammatory activities of *S. unitus* were as follows: acetone (*t* = 0.006), chloroform (*t* = 0.008), methanol (*t* = 0.02), and petroleum ether (*t* = 0.004) extracts. Correlation values between the metabolites’ concentration and anti-inflammatory activity of *S. unitus* was calculated and presented in Table 7.

### 2.6. α-Amylase Activity

In *S. unitus* extract, the highest percentage of α-amylase activity (80%) was observed for the petroleum ether extract (25 µg/mL) and the lowest activity (10 %) for the methanol extract (5 µg/mL), as shown in Figure 3. The mean difference was considered statistically significant at the 0.05 level. The *t*-values for the extracts and antidiabetic activities of *S. unitus* were as follows: acetone (*t* = 0.026), chloroform (*t* = 0.009), methanol (*t* = 0.004), and petroleum ether (*t* = 0.005) extracts. Correlation values between the metabolites’ concentration and α-amylase activity of *S. unitus* was calculated and presented in Table 7.

## 3. Discussion

Primary and secondary metabolites are widely found at different levels in many medicinal plants [27]. Of them, secondary metabolites (SMs) serve as defense compounds against biotic and abiotic components [27]. The available literature confirmed their wide array of biological and pharmacological properties [28,29,30]. Indeed, plant crude extracts and its isolated fractions have been increasingly employed to treat infections, health disorders, or diseases [28,29,30]. Plants used in phytotherapy are usually rich in bioactive molecules, with phenolic compounds (flavonoids, catechins, phenylpropanoids, rosmarinic acid, tannins, polyketides), terpenoids (mono- and sesquiterpenes, iridoids, saponins), and polysaccharides being the most abundant ones [28]. In general, terpenes show cytotoxic activities against a wide range of organisms, ranging from bacteria and fungi to insects and vertebrates, and have been widely used in herbal medicine against infections [28,29,30,31]. Phenolics are accountable for broad pharmacological properties, viz., antioxidant, anti-inflammatory, sedating, wound healing, and antimicrobial and antiviral activities [28,29,30]. Indeed, flavonoids are the active ingredients of many herbal medicines [28,29,30]. Tannins are strong antioxidant, anti-inflammatory, antidiarrheal, cytotoxic, antiparasitic, antibacterial, antifungal, and antiviral agents. 

The present study aims to address the *S. unitus* chemical composition, being stated as an interesting abundance of tannins, phenolic, saponins, triterpenoids, sterols, and glycosides, with varied percentages. Data obtained in this study suggest that *S. unitus* may possess various biological and pharmacological properties, such as antioxidant, anti-inflammatory, antidiabetic, and cytotoxic activities. To confirm the biological properties of *S. unitus* secondary metabolites and to identify their medicinal properties, the antioxidant capacity toward free radical propagation and antidiabetic, anti-inflammatory, and toxic potential against *Artemia salina* were addressed.

Brine shrimp lethality bioassay (BSLB) was employed to determine toxicity through the estimation of medium lethal concentration (LC_50_), which has been reported for a series of toxins and plant extracts. Several naturally extracted products with LC_50_ values <1000 μg/mL using BSLB are known to contain physiologically active principles [32]. BSLB and other in vivo lethality tests have been successively employed for bioassay guide fractionation of active cytotoxic and antitumor agents [33,34]. BSLB is considered to be very useful in determining various biological activities, such as cytotoxic, phototoxic, pesticidal, trypanocidal, enzyme inhibition, and ion regulation activities [35], and it can also be extrapolated for cell-line toxicity and antitumor activity [36]. These findings clearly indicated that the studied extracts of *S. unitus* possess physiologically active principles.

Reactive oxygen species (ROS) can react with important cell macromolecules, such as proteins, lipids, and nucleic acids [37]. As a consequence, an overdose of ROS may lead to several, mostly chronic, health disorders, such as diabetes, metabolic syndrome, cardiovascular disease, and even cancer [38]. ROS may also influence the aging process [39]. Many phenolics, terpenoids with conjugated double bonds, and ascorbic acid are able to inhibit ROS and other oxygen radicals [40]. Many herbal drugs and products from phenolic-rich plants may, therefore, exhibit antioxidant activity. Phenolic compounds, such as flavonoids, phenolic acids, and tannins, exert remarkable anti-inflammatory, anticarcinogenic, antiatherosclerotic, and other biological effects that may be related to their antioxidant activities [41]. 

The antioxidant assays used in this study measured the oxidative products at the early and final stages of oxidation. The results from these in vitro experiments, including ABTS radical monocation scavenging, DPPH radical scavenging, phosphomolybdenum assay, and total polyphenol, total flavonoid, total triterpenoids, total steroids and total tannin contents, revealed that *S. unitus* phytochemicals might have significant effects as antioxidant agents. However, the quantity of polyphenols and flavonoids found in *S. unitus* extracts were not directly related to their antioxidant activities, meaning that the additive role of phytochemicals might contribute significantly to the potent antioxidant activity observed. Hence, *S. unitus* could be used as an easily accessible source of natural antioxidants in both the pharmaceutical and medical industries. 

On the other hand, the human RBC membrane stabilization (HRBC) method has been employed for the determination of anti-inflammatory effects [24,25]. In this study, the results obtained indicated that the various *S. unitus* extracts, at various doses, exerted significant anti-inflammatory effects. Both qualitative and quantitative analyses confirmed the occurrence of triterpenoids in the crude extract of *S. unitus*. The phytochemical analysis results suggest that secondary metabolites present in the crude extract of *S. unitus* may be responsible for the anti-inflammatory activity, namely, its content in triterpenes. 

When looking at *S. unitus* antidiabetic potential, the findings of this study revealed that petroleum ether extract at 25 µg/mL produced 75.92% α-amylase inhibition, while the standard drug, acarbose, showed 90.91% inhibition at 100 µg/mL. Previous studies have reported that phenolic compounds inhibit the α-amylase enzyme by modulating the enzymatic breakdown of carbohydrates [42]. Thus, it can be concluded from the abovementioned results that the α-amylase inhibitory potential observed to the tested extracts of *S. unitus* might be attributed to their phenolic content. Thus, further analysis of the active compounds should be carried out, as they may provide more information on the role of polyphenols and flavonoids present in *S. unitus* towards disease management, where there is an increase in oxidative stress.

## 4. Materials and Methods

### 4.1. Preparation of Plant Extract

The matured sporophytes of *S. unitus* were collected from their natural habitats (Palni Hills, Kodaikannal, Tamil Nadu, India). The specimens were identified by Dr. M. Johnson (Fellow of the Indian Fern Society, India) using the Flora of Pteridophytes. The collected sporophytes (aerial parts) were shade dried at room temperature for 15 days. The dried materials were powdered using a mechanical grinder (Butterfly Mixer Grinder, Chennai, India) for 30 min. 

The powdered materials of *S. unitus* were sequentially extracted with (1: 6 ratio *w/v*) petroleum ether (45 °C), chloroform (55 °C), acetone (52 °C), and methanol (75 °C) by using a Soxhlet extractor (LabMen, Mumbai, India) for 8 h, at a temperature not exceeding the boiling point of the solvent. Hi Media (Mumbai, India) chemicals and solvents were employed for the extraction and biological analysis. The quantification of metabolites and estimation of biological activities were measured using a Shimadzu UV–vis spectrophotometer, Japan.

### 4.2. Preliminary Phytochemical Analysis

The various extracts of *S. unitus* (SUE) were tested for the presence of phenols, tannins, steroids, cardiac glycosides, saponins, amino acids, terpenoids, and flavonoids using the standard method (Table 8) described by Harborne [42].

### 4.3. Quantification of Total Phenolic

Total phenolic content in *S. Unitus* extracts was determined following the method described by Siddhuraju and Becker [43]. Briefly, 100 µg/mL of various *S. unitus* extracts were taken in test tubes and made up to the volume of 1 mL with distilled water. Then, 0.5 mL of 1 N Folin–Ciocalteu reagent (1:1 with water) and 2.5 mL of sodium carbonate solution (20%) were added sequentially in each tube. The reaction mixture was vortexed, and the test tubes were kept in the dark for 40 min, and the absorbance was recorded at 725 nm against blanks. The analysis was performed in triplicate, and the results were expressed as mg GAE/g DW.

### 4.4. Quantification of Total Tannins

To quantify the total tannins, 100 mg of polyvinyl polypyrrolidine (PVPP) were put in an Eppendorf, and the volume was adjusted to 1 mL with distilled water. Then, 100 µg/mL of the various *S. unitus* extracts were added and incubated at 4 °C for 4 h. PVPP precipitates the tannin content. Then, the solution was centrifuged at 4000 rpm for 10 min, and the supernatant was collected. The reaction mixture (supernatant) was made up to the known volume with distilled water. Then, 0.5 mL of 1 N Folin–Ciocalteu reagent (1:1 with water) and 2.5 mL of sodium carbonate solution (20%) were added sequentially in each tube. Soon after vortexing the reaction mixture, the test tubes were placed in the dark for 40 min, and the absorbance was recorded at 725 nm against blanks. The analysis was performed in triplicate, and the results were expressed as mg GAE/g DW [43].

### 4.5. Quantification of Total Flavonoids

The flavonoid contents of *S. unitus* extract were quantified as it acts as a major antioxidant in plants, reducing oxidative stress, as described by Zhishen et al. [44]. Initially, 100 µg/mL of various *S. unitus* extracts were taken in different test tubes. To each extract, 2 mL of distilled water was added. Then, 150 µL of 5% NaNO_2_ was added to all the test tubes, followed by incubation at room temperature for 6 min. After incubation, 150 µL of AlCl_3_ (10%) was added to all the test tubes, including the blank. All test tubes were then incubated for 6 min at room temperature. Afterwards, 2 mL of 4% NaOH was added, which was made up to 5 mL using distilled water. The reaction mixtures were vortexed well and allowed to stand for 15 min at room temperature. The appearance of pink color was recorded and measured spectrophotometrically at 510 nm. The amount of flavonoids was calculated in mg RE/g DW.

### 4.6. Quantification of Total Sterols

Total sterols content was determined using the modified Liebermann–Burchard colorimetric assay [45]. The Liebermann–Burchard reagent was prepared by adding 1.25 mL of concentrated H_2_SO_4_ to 50 mL of acetic anhydride. The Liebermann–Burchard reagent (2 mL) was mixed with 100 µg/mL of the various *S. unitus* extracts prepared in chloroform at different concentrations (50–200 µg/mL), stirred for 1 min and incubated at room temperature (26 °C) for 13 min. The absorbance was measured at 650 nm, using cholesterol as standard. The total sterols content was expressed as mg/g cholesterol equivalent DW.

### 4.7. Quantification of Total Terpenoids

The terpenoids content of *S. unitus* extracts was quantified by taking 1 mL of 2% vanillin and adding 100 µg/mL of the various *S. unitus* extracts prepared in methanol and agitating the mixtures in an ice bath for 10 min. After agitation, all test tubes were incubated at 60 º C for 20 min in a water bath. The test tubes were then cooled at 25 º C for 5 min. All test tubes were read at 608 nm against blanks [46].

### 4.8. Toxicity Analysis Using Brine Shrimp Lethality Bioassay

The hatched brine shrimp’s nauplii were employed for BSLB. Five different concentrations, viz. 1, 2, 3, 4, and 5 mg/mL of *S. unitus* extracts were employed. Distilled water was employed as control. With the help of a Pasteur pipette, 10 living shrimp nauplii were dropped into each test tube [47]. 

#### Counting of Nauplii

After 24 h, the tubes were inspected using a magnifying glass, and the number of survived nauplii in each tube was counted, and the LC_50_ (with a 95% confidence limit), LCL, and UCL were calculated.

### 4.9. Antioxidant Activity

To know the antioxidant potential of *S. unitus* extracts, DPPH radical scavenging activity [48], phosphomolybdenum assay [49], superoxide radical scavenging activity [50], and ABTS assays [51] at varying concentrations were used.

#### 4.9.1. DPPH Radical Scavenging Activity

The *S. unitus* extracts at various concentrations were taken, and the volume was adjusted to 100 µL with methanol. About 5 mL of a 0.1 mM methanol solution of DPPH was added to the aliquots of samples and standards (Plumbagin) and shaken vigorously. Negative control was prepared by adding 100 µL of methanol in 5 mL of a 0.1 mM methanol solution of DPPH. The test tubes were allowed to stand for 20 min at 27 ˚C. The absorbance of samples was measured at 517 nm against the blank (methanol). Radical scavenging activity of the samples was expressed as IC_50_, which corresponds to the sample concentration required to inhibit 50% of DPPH concentration.

#### 4.9.2. Phosphomolybdenum Assay

The *S. unitus* extracts at varied concentrations (in 1 mM dimethyl sulfoxide) were combined with 1 mL of reagent solution (0.6 M sulfuric acid, 28 mM ammonium molybdenum) in a 4 mL vial. The vials were capped and incubated in a water bath at 95˚ C for 90 min. After the samples were cooled at room temperature, the absorbance of the mixture was measured at 695 nm against the blank. The results were expressed as mean values, as g of ascorbic acid (AA) equivalents/100 g extract.

#### 4.9.3. Superoxide Radical Scavenging Activity

The various *S. unitus* extracts were prepared using sodium phosphate buffer (pH 6). Superoxide radical scavenging was determined from absorption spectra at 590 nm. Briefly, 20 mg riboflavin, 12 mM EDTA, and 0.1 mg NBT were prepared in sodium phosphate buffer (3 mL), and 3 mL of reagent solution illuminated for 90 sec. For each concentration, a ascorbic acid blank sample was used for background subtraction. The percentage inhibition activity was calculated using the following formula: % of scavenging activity = [(Control OD − Sample OD)/Control OD] × 100.

#### 4.9.4. ABTS Assay

The radical scavenging activity of *S. unitus* extracts was analyzed by the 2,2-azino-bis (3-ethylbenzothiazoline-6-sulfonic acid) assay (ABTS). ABTS was produced by reacting 7 mM ABTS aqueous solution with 2.4 mM potassium persulfate. This mixture was kept at room temperature for 12–16 h. Prior to assay, this solution was diluted in ethanol (about 1.89 *v/v*) and equilibrated at 30 °C to give an absorbance at 734 nm of 0.70 ± 0.02. After the addition of 1 mL of diluted ABTS solution to 10 µL of *S. unitus* extract or Trolox standard (final concentration 0–15 µM) in ethanol, absorbance was measured at 30 °C exactly 30 min after the initial mixing. An appropriate solvent blank was also run. Triplicate analyses were made at each dilution of the standard, and the inhibition percentage was evaluated at 734 nm. 

### 4.10. Anti-inflammatory Activity

To reveal the anti-inflammatory activity of *S. unitus* extracts, the RBC membrane stabilization test was adopted [52]. The reaction mixture (2 mL) consisted of 1 mL of *S. unitus* extract and 1 mL of 10% RBC suspension; only saline was added to the control test tube. Aspirin was taken as a standard drug. The reaction mixtures were kept in a water bath at 56 °C for 30 min. After the incubation, the reaction mixtures were cooled to room temperature. The reaction mixture was centrifuged at 2500 rpm for 5 min, the supernatants were collected, and the absorbance measured at 560 nm. The experiment was performed in triplicate for all the test samples. Percent RBC membrane stabilization activity was calculated using the following formula: % inhibition = Control ABS - Sample ABS / Control ABS * 100.

### 4.11. Antidiabetic Activity

To determine the antidiabetic potential of *S. unitus*, the in vitro α-amylase inhibitory assay was employed. Briefly, 5, 10, 15, 20, and 25 (µg/mL) of *S. unitus* extracts were taken in a tube, and 1.2 mL of starch in phosphate buffer (pH 6.9) containing 6.7 mM of sodium chloride was added. Porcine pancreatic amylase (600 µL) was added to the mixture and incubated at 37 °C. From the above mixture, 600 µL was taken, and 300 µL of DNSA was added and kept in a boiling water bath for 15 min. Distilled water (3.1 mL) was added, and absorbance was measured at 540 nm. For each concentration, blank tubes were prepared by replacing the enzyme solution with 600 μL of distilled water. Control, representing 100% enzyme activity, was prepared in a similar manner, without extract. The experiments were replicated thrice using the same protocol. 

The α-amylase inhibitory activity was calculated using the formula: α-amylase inhibitory activity = (Ac+) − (Ac−) − (As − Ab)/(Ac+) − (Ac −) × 100, where Ac+, Ac−, As, and Ab are defined as the absorbance of 100% enzyme activity (only solvent with enzyme), 0% enzyme activity (only solvent without enzyme), a test sample (with enzyme), and a blank (test sample without enzyme), respectively.

### 4.12. Statistical Analysis

To validate the observed results, the statistical analysis was performed using the Statistical Package for the Social Sciences (SPSS, IBM Corp. USA) software, version 21. Pearson’s correlation test was performed between the metabolite’s concentration and biological activities. The correlation was significant at the 0.05 level (2-tailed). A *t*-test was performed between the metabolites’ concentration and biological activities.

## 5. Conclusions

In this study, *S. unitus* revealed a great ability to fight against harmful free radicals and seems to be a broad-spectrum antioxidant agent. Toxicity of *S. unitus* in terms of the brine shrimp lethal effect was found to be most effective for isolated biogenic compounds from the plant extract. For this reason, further studies should be carried out to isolate and to identify the responsible individual compounds for the antioxidant, anti-inflammatory, and antidiabetic effects.

## Figures and Tables

**Figure 1 antibiotics-09-00333-f001:**
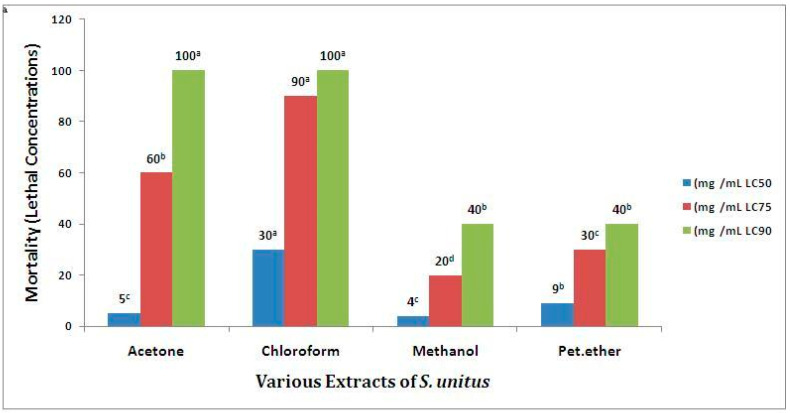
Toxicity of *S. unitus* (LC_50_, LC_75_, and LC_90_ in mg/mL). The mean difference was considered statistically significant at the 0.05 level. Samples with similar superscripts were significantly similar (*p* > 0.05). The mean values were rounded off to one significant figure (n = 20). The albaha subset a, b, c and d are significantly different.

**Figure 2 antibiotics-09-00333-f002:**
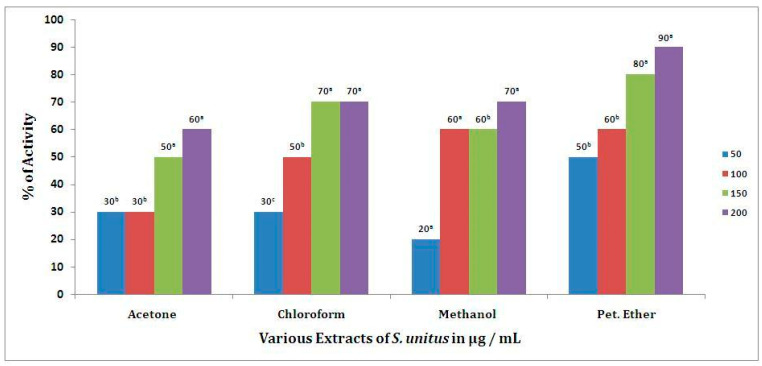
Anti-inflammatory activity of *S. unitus.* The mean difference was considered statistically significant at the 0.05 level. Samples with similar superscripts were significantly similar (*p* > 0.05). The mean values were rounded off to one significant figure (n = 12). The albaha subset a, b and c are significantly different.

**Figure 3 antibiotics-09-00333-f003:**
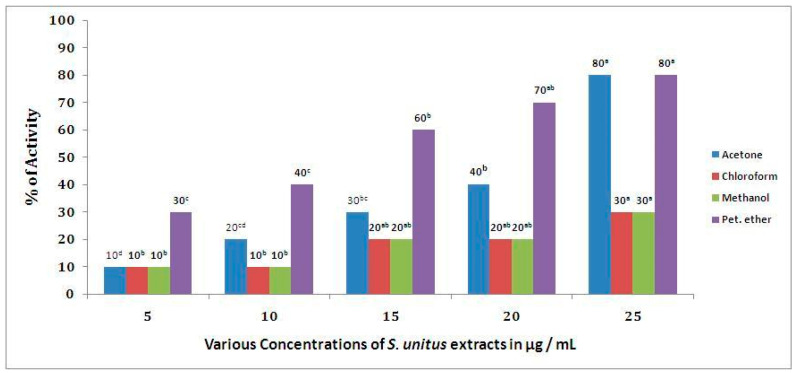
α-Amylase activity of *S. unitus.* The mean difference was considered statistically significant at the 0.05 level. Samples with similar superscripts were significantly similar (*p* > 0.05). The mean values were rounded off to one significant figure (n = 15). The albaha subset a, b, c and d are significantly different.

**Table 1 antibiotics-09-00333-t001:** Preliminary phytochemical studies of *S. unitus.*

Metabolites	Acetone	Chloroform	Methanol	P. ether	Total
Steroids	+	+	+	+	4
Alkaloids	-	-	-	-	0
Phenolic groups	+	+	+	+	4
Cardiac glycosides	+	+	+	+	4
Flavonoids	+	+	+	+	4
Saponins	+	+	+	+	4
Tannins	+	+	+	+	4
Terpenoids	+	+	+	+	4
Total	7	7	7	7	28

**Table 2 antibiotics-09-00333-t002:** Total phenols, steroids, tannins, flavonoids, and terpenoids of *S. unitus.*

Metabolites	Acetone	Chloroform	Methanol	P. ether
Total phenolic (mg GAE/g DW)	50 ± 5 ^c^	90 ± 20 ^b^	300 ± 10 ^a^	60 ± 20 ^bc^
Total tannins (mg GAE/g DW)	30 ± 4 ^b^	10 ± 1 ^d^	50 ± 2 ^a^	20 ± 2 ^c^
Total steroids (mg/g)	0.2 ± 0.01 ^a^	0.2 ± 0.03 ^a^	0.09 ± 0.01 ^b^	0.1 ± 0.04 ^b^
Total flavonoids (mg RE/g)	30 ± 10 ^b^	60 ± 20 ^b^	200 ± 40 ^a^	70 ± 4 ^b^
Total terpenoids (mg/g)	200 ± 8 ^a^	200 ± 4 ^a^	200 ± 9 ^a^	200 ± 8 ^a^

The mean difference was considered statistically significant at the 0.05 level. Samples with similar superscripts were significantly similar (*p* > 0.05). The mean values were rounded off to one significant figure (n = 3). The albaha subset a, b, c and d are significantly different.

**Table 3 antibiotics-09-00333-t003:** DPPH scavenging activity of *S. unitus.*

Concentration (µg/mL)	% of Activity			
	Acetone	Chloroform	Methanol	P. Ether
50	60 ± 0.8 ^c^	40 ± 0.6 ^c^	80 ± 0.5 ^a^	50 ± 0.3 ^b^
100	70 ± 0.8 ^b^	50 ± 0.8 ^b^	80 ± 0.6 ^a^	50 ± 0.4 ^b^
150	70 ± 1 ^b^	60 ± 0.8 ^a^	80 ± 0.5 ^a^	50 ± 0.6 ^b^
200	80 ± 0.9 ^a^	60 ± 0.9 ^a^	80 ± 0.7 ^a^	100 ± 0.8 ^a^
IC_50_	100	100	90	100

The mean difference was considered statistically significant at the 0.05 level. Samples with similar superscripts were significantly similar (*p* > 0.05). The mean values were rounded off to one significant figure (n = 12). The albaha subset a, b and c are significantly different.

**Table 4 antibiotics-09-00333-t004:** Phosphomolybdenum assay of *S. unitus* (TAC-mg AA/g).

Concentration (µg/mL)	Acetone	Chloroform	Methanol	P. ether
**50**	1 ± 0.1 ^b^	1 ± 0.03 ^b^	0.9 ± 0.5 ^b^	1 ± 0.1 ^b^
**100**	9 ± 0.5 ^a^	1 ± 0.5 ^b^	1 ± 0.04 ^b^	1 ± 0.3 ^b^
**150**	10 ± 2 ^a^	1 ± 0.4 ^b^	1 ± 0.1 ^b^	1 ± 0.3 ^b^
**200**	10 ± 2 ^a^	2 ± 0.4 ^a^	2 ± 0.4 ^a^	3 ± 0.5 ^a^

The mean difference was considered statistically significant at the 0.05 level. Samples with similar superscripts were significantly similar (*p* > 0.05). The mean values were rounded off to one significant figure (n = 12). The albaha subset a and b are significantly different.

**Table 5 antibiotics-09-00333-t005:** Superoxide assay (% of activity) of *S. unitus.*

Concentration (µg/mL)	Acetone	Chloroform	Methanol	P. ether
**50**	70 ± 0.9 ^b^	80 ± 0.6 ^b^	100 ± 0.3 ^a^	70 ± 0.2 ^c^
**100**	70 ± 0.6 ^b^	90 ± 0.8 ^a^	100 ± 0.2 ^a^	80 ± 0.2 ^b^
**150**	90 ± 0.6 ^a^	90 ± 0.5 ^a^	100 ± 0.2 ^a^	80 ± 0.3 ^b^
**200**	90 ± 0.8 ^a^	90 ± 0.6 ^a^	100 ± 0.2 ^a^	90 ± 0.2 ^a^

The mean difference was considered statistically significant at the 0.05 level. Samples with similar superscripts were significantly similar (*p* > 0.05). The mean values were rounded off to one significant figure (n = 12). The albaha subset a, b and c are significantly different.

**Table 6 antibiotics-09-00333-t006:** 2,2-Azino-bis (3-ethylbenzothiazoline-6-sulfonic acid) (ABTS) assay of *S. unitus (*µmol/g).

Concentration (µg/mL)	Methanol	Chloroform	Pet. ether	Acetone
**50**	700 ± 2 ^c^	700 ± 3 ^d^	700 ± 0.1 ^c^	700 ± 2 ^c^
**100**	900 ± 5 ^bc^	900 ± 3 ^c^	900 ± 10 ^bc^	900 ± 1 ^bc^
**150**	1000 ± 40 ^b^	1000 ± 7 ^b^	1000 ± 20 ^b^	1000 ± 100 ^b^
**200**	2000 ± 200 ^a^	3000 ± 40 ^a^	2000 ± 70 ^a^	2000 ± 600 ^a^

The mean difference was considered statistically significant at the 0.05 level. Samples with similar superscripts were significantly similar (*p* > 0.05). The mean values were rounded off to one significant figure (n = 12). The albaha subset a, b and c are significantly different.

**Table 7 antibiotics-09-00333-t007:** Correlation values between the metabolites’ concentration and antioxidant, anti-inflammatory, and antidiabetic activities of *S. unitus* extracts.

Activities	Acetone	Methanol	Chloroform	P. ether
**DPPH**	0.967 *	0.949 *	0.927 *	0.848 *
**SOD**	0.989 *	0.981 *	0.879	0.971 *
**Anti-inflammatory**	0.958 *	0.838 *	0.973 *	0.995 *
**Anti-diabetic**	0.915 *	0.981 *	0.872 *	0.983 *

* Correlation was significant at the 0.05 level (2-tailed).

**Table 8 antibiotics-09-00333-t008:** Preliminary Phytochemical Analysis of *S. unitus*

Test	Observation	Inference
1 mL of SUE + chloroform + 3 drops of acetic anhydride + 1 drop of con. sulfuric acid.	Purple color develops changing blue (or) green	Presence of steroids
1 mL of SUE + 2N HCL + 2 drops of Mayer’s reagent	White turbidity (or) precipitate	Presence of alkaloids
1 mL of SUE + 1 mL of 80% alcohol + 1 drop of ferric chloride + 2 mL of glacial acetic acid	Intense color develops	Presence of phenolic groups
1 mL of SUE + ferric chloride + 2 mL of glacial acetic acid + H_2_SO_4_	Brown, violet, and greenish	Presence of cardiac glycosides
1 mL of SUE + dis. H_2_O	Foamy lather develops	Presence of saponins
1 mL of SUE + water + lead acetate	White precipitate	Presence of tannins
1 mL of SUE + trace tin metal + 2 mL thionyl chloride	Pink color	Presence of terpenoids
1 mL of SUE + 5% NaNO_2_ + AlCl_3_ + NaOH	Pink color	Presence of flavonoids

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
