# Peer review of "In Vitro Toxicity, Antioxidant, Anti-Inflammatory, and Antidiabetic Potential of *Sphaerostephanos unitus* (L.) Holttum"

_antibiotics, 2020, doi:10.3390/antibiotics9060333_

Round 1

Reviewer 1 Report

This manuscript descript the different solvent extract for thier in vitro assay. The authors should explain why using different solvent to extract the material rather than using one solvent and then partitioned by different solvent. And my other comments are list below:

  1. The result of table 1 and 2 are not reasonable. In table 1, steroids are not existed in acetone, chloroform, and methanol. But in table 2, steroids can be tested and quantitated? As well as phenolic groups, flavonoids, and terpenoids.
  2. In table 2, the data are contradictory. Cause tannins and flavonoids are also phenolic compounds. Why the content of phenolics are less than tannins and flavonoids?
  3. In 4.2 preliminary phytochemical analysis, the detail method should recorded. It is not enough to cite reference.
  4. In 4.4 and 4.7, the methods should cite references.

Author Response

This manuscript descripts the different solvent extract for thier in vitro assay. The authors should explain why using different solvent to extract the material rather than using one solvent and then partitioned by different solvent. And my other comments are list below:

  1. The result of table 1 and 2 are not reasonable. In table 1, steroids are not existed in acetone, chloroform, and methanol. But in table 2, steroids can be tested and quantitated? As well as phenolic groups, flavonoids, and terpenoids.

Ans: The reviewers queries were carefully considered and the qualitative analysis was repeated. The obtained results were recorded.

  1. In table 2, the data are contradictory. Cause tannins and flavonoids are also phenolic compounds. Why the content of phenolics are less than tannins and flavonoids?

Ans: The typographical errors were rectified.

  1. In 4.2 preliminary phytochemical analysis, the detail method should recorded. It is not enough to cite reference.

Ans: The detailed procedure was added.

  1. In 4.4 and 4.7, the methods should cite references.

Ans: The references were included. 

Reviewer 2 Report

The study by Johnson and co-workers deals with the phytochemical content and biological activities of different extracts obtained from the fern Sphaerostephanos subtruncatus. In particular, the presence of steroids, alkaloids, phenolics and isoprenoids in different plant extracts was evaluated, and toxicity, antioxidant, anti-diabetic and anti-inflammatory tests were carried out.

Overall, the results of this study suggest that S. subtruncatus could have interesting applications as medicinal plant. In my opinion the study is interesting, the manuscript is well written, and it is suitable for publication in Antibiotics. However, I suggest making some small changes to the manuscript that could improve its quality.

  • The first time a plant species is mentioned, the authority should be reported.

  • Both in the figures and in the tables the statistical significance should be reported.

  • The figure and table captions should be self-explanatory, then I recommend improving them.

  • As for Fig. 1, 2 and 3, I recommend using histograms, because line graphs are more suitable for describing phenomena that take place over time.

  • In Materials and methods section, the part of the plant used to obtain the extracts should be reported, as the phytochemical profile varies greatly in different plant organs.

  • In Materials and methods section, the reagent suppliers and the instrument technical features should be reported.

Other suggested editorial changes are reported in the attached pdf file.

Author Response

The study by Johnson and co-workers deals with the phytochemical content and biological activities of different extracts obtained from the fern Sphaerostephanos unitus. In particular, the presence of steroids, alkaloids, phenolics and isoprenoids in different plant extracts was evaluated, and toxicity, antioxidant, anti-diabetic and anti-inflammatory tests were carried out.

 Overall, the results of this study suggest that S. unitus could have interesting applications as medicinal plant. In my opinion the study is interesting, the manuscript is well written, and it issuitable for publication in Antibiotics. However, I suggest making some small changes to the manuscript that could improve its quality.

  • The first time a plant species is mentioned, the authority should be reported.

Ans: The authority name is included in the Materials section as follows The specimens were identified by Dr. M. Johnson (Fellow of Indian Fern Society, India) using the Flora of Pteridophytes.

  • Both in the figures and in the tables the statistical significance should be reported.

Ans: As per the reviewers suggestion the test of significance results were included

  • The figure and table captions should be self-explanatory, then I recommend improving them.

 Ans: As per the reviewers suggestion the captions were modified.

  • As for Fig. 1, 2 and3, I recommend using histograms, because linegraphs are more suitable for describing phenomena that takeplace over time.

Ans: As per the reviewers suggestion the graphs were modified.

  • In Materials and methods section, the part of the plant used to obtain the extracts should be reported, as the phytochemical profile varies greatly in different plant organs.

Ans: As per the reviewers suggestion the part of the plant used for extraction was reported.

  • In Materials and methods section, the reagent suppliers and the instrument technical features should be reported.

Ans: As per the reviewers suggestion the reagent supplier and the instruments were included. The Hi Media (Mumbai, India) chemicals and solvents are employed for the extraction and biological analysis. The quantification of metabolites and estimation of biological activities was measured using Shimadzu UV-Vis spectrophotometer, Japan.

Other suggested editorial changes are reported in the attached pdf file.

Ans: As per the reviewers suggestion we have modified the contents.

Reviewer 3 Report

The present study describes the qualitative and quantitative secondary  metabolites profile (total phenolics, tannins, steroids, flavonoids and terpenoids obtained from the various extracts) of Sphaerostephanos unitus and their toxicity, antioxidant, anti-diabetic and anti-inflammatory potentials.

The paper is interesting and in the aim of the journal, but it requires major revision because needs additional work and details in terms of accuracy and precision of writing in abstract, keywords, introduction, materials and methods, results, discussion and conclusions.

The statistical analysis section is missing, please add statistical analysis output  for each treatment to evaluate the statistical significance of differences. Also, there is a lot of confusion in the use of standards, e.g. gallic acid to quantify flavonoids, rutin for DPPH….

 My specific comments and suggestions for improving the paper are:

Abstract

According to authors guideline “The abstract should be a single paragraph and should follow the style of structured abstracts, but without headings”

Line 15, please delete Background

Line 19, please delete Methods

Line 21, please delete Results

Line 20, please delete Conclusion

Line 31, please put individual compounds instead of constituents

Keywords

I suggest changing the keywords that are already present in the title (as  Sphaerostephanos unitus; Antioxidant; Anti-diabetic; Anti-inflammatory; Toxicity.

Introduction

Line 45-46, please summarize….as many authors 5-14

Results

Line 68, why  5/10??? Total metabolites are 8…..

Line 75, please insert phenolic instead of phenolics

Line 76, please define the abbreviation GAE

Table 2, it is necessary to report the statistical analysis output  for each phytochemicals from the various extracts, to evaluate the statistical significance of differences

Table 3,4,5,6 please add statistical analysis output  for each treatment to evaluate the statistical significance of differences

Figure 1,2,3 please add statistical analysis output  for each treatment to evaluate the statistical significance of differences

Table 6, please in the caption after ABTS assay add parameters of quantification and specified better in materials and methods. How are the results expressed?  Is it µmol/g?

Discussion

Line 158, please insert reference

Line 191-196, these sentences are results….please move to the results and insert a table

Line 207-209, as above

Line 216-218 as above

Materials and methods

Line 228-230, please insert a reference

Line 239, Birefly ???? (typing error) perhaps Briefly

Line 244, please add DW after mg GAE/g

Line 247-249, please insert a reference

Line 266, please specified better this sentence“The amount of flavonoids was calculated in GAE” Why do you use GAE (galllic acid) instead of catechin or rutin that are flavonoids? In addition are mg/g?

Line 278-281, please add a reference

Line 301, Why do you use rutin instead of trolox

Line 324, TEAC???? Perhaps ABTS

Line 324 , Why do you insert TEAC? TEAC means Tolox Equivalent Antioxidant Capacity

Line 331, please specified better this sentence“The inhibition percentage was plotted against Trolox concentration” Is it IC50?………and where is the value of this? In the results are missing….

Line 332-340, please insert a sub heading Anti-inflammatory activity

Line 332, please define the abbreviation RBC

Line 344, please insert a reference

Line 347, please define the abbreviation DNSA

Please add the statistical analysis section

Conclusion

Line 362, please add individual before compound

Author Response

The present study describes the qualitative and quantitative secondary  metabolites profile (total phenolics, tannins, steroids, flavonoids and terpenoids obtained from the various extracts) of Sphaerostephanos unitus and their toxicity, antioxidant, anti-diabetic and anti-inflammatory potentials.

The paper is interesting and in the aim of the journal, but it requires major revision because needs additional work and details in terms of accuracy and precision of writing in abstract, keywords, introduction, materials and methods, results, discussion and conclusions.

The statistical analysis section is missing, please add statistical analysis output for each treatment to evaluate the statistical significance of differences. Also, thereis a lot of confusion in the use of standards, e.g. gallic acid to quantify flavonoids, rutin for DPPH….

My specific comments and suggestions for improving the paper are:

Abstract

According to authors guideline “The abstract shouldbe a single paragraph and should follow the style of structured abstracts, but without headings”

Line 15, please delete Background

Line 19, please delete Methods

Line 21, please delete Results

Line 20, please delete Conclusion

Line 31, please put individual compounds instead of constituents

ANS: As per the reviewers’ suggestion, the abstract was converted. Constituents was replaced by compounds.

Keywords

I suggest changing the keywords that are already present in the title (as  Sphaerostephanos unitus; Antioxidant; Anti-diabetic; Anti-inflammatory; Toxicity.

ANS: As per the reviewers’ suggestion, the keywords were modified.

Introduction

Line 45-46, please summarize….as manyauthors 5-14

ANS: As per the reviewers’ suggestion, the sentence construction was modified.

Results

Line 68, why 5/10??? Total metabolites are 8…..

ANS: As per the reviewers’ suggestion, the experiment was repeated, the presence of metabolites were reconfirmed and data modified.

Line 75, please insert phenolic instead of phenolics

ANS: As per the reviewers’ suggestion, the phenolics were used in the place of phenolics.

Line 76, please define the abbreviation GAE

ANS: GAE – Gallic Acid Equivalent was included

Table 2, it is necessary to report the statistical analysis output for each phytochemicals from the various extracts, to evaluate the statistical significance of differences

ANS: The average and Standard deviation for the metabolites are calculated and presented

Table3,4,5,6 please add statistical analysis output for each treatment to evaluate the statistical significance of differences

ANS: The average and Standard deviation for the metabolites was calculated and presented. Ans: The one sample t-Test was performed and the results were incorporated.

Figure 1,2,3 please add statistical analysis output for each treatment to evaluate the statistical significance of differences

Ans: The one sample t-Test was performed and the results were incorporated.

Table 6, please in the caption after ABTS assay add parameters of quantification and specified better in materials and methods. How are the results expressed? Is it µmol/g?

ANS: As per reviewers’ suggestion, the quantification of ABTs assays was mentioned as µmol/g.

Discussion

Line 158, please insert reference

Ans: As per the reviewers’ suggestion, the reference was included

Line 191-196, these sentences are results….please move to the results and insert a table

Ans: As per the reviewers’ suggestion, table 7 was included.

Line 207-209, as above

Ans: As per the reviewers’ suggestion, table 7 was included.

Line 216-218 as above

Ans: As per the reviewers’ suggestion, table 7 was included.

Materialsandmethods

Line 228-230, please insert a reference

Line 239, Birefly ???? (typingerror) perhapsBriefly

Ans: As per the reviewers’ suggestion, the typographical error was corrected.

Line 244, please add DW after mg GAE/g

Ans: As per the reviewers’ suggestion, the DW was included after GAE.

Line 247-249, please insert a reference

Ans: Reference included

Line 266, please specified better thi ssentence“The amount of flavonoids was calculated in GAE” Why do you use GAE (galllicacid) instead of catechin or rutint hat are flavonoids? In addition are mg/g?

Ans: The amount of flavonoids was calculated in mg RE/g mistakenly it denoted as GAE.

Line 278-281, please add a reference

Ans:Harborne J. B. Phytochemical Methods. London, UK: Chapman & Hall; 1973

Line 301, Why do you use rutin instead of trolox

Ans: We have performed with plumbagin and Trolox only. In the future we will employ the rutin in the place of Trolox.

Line 324, TEAC????Perhaps ABTS

Ans: The ABTS is replaced in the place of TEAC.

Line324 ,Why do youinsert TEAC? TEAC meansToloxEquivalentAntioxidantCapacity

Ans: We removed the TEAC.

Line 331, please specified better this sentence “The inhibition percentage was plotted against Trolox concentration” Is it IC50?………and where is the value of this? In the results are missing….

Ans: To find the factorial values we used the Trolox.

Line 332-340, please insert a sub heading Anti-inflammatory activity

Ans: As per the reviewers’ suggestion the sub heading was included.

Line 332, please define the abbreviation RBC

Ans: As per the reviewers’ suggestion the RBS was defined in Abbreviations.

Line 344, please insert a reference

Ans: Ref. 52 was included

Line 347, please define the abbreviation DNSA

Ans: As per the reviewers’ suggestion the DNSA was defined in Abbreviations.

Please add the statistical analysis section

Ans: As per the reviewers’ suggestion the statistical methods were included in the Materials and Methods section.

Conclusion

Line 362, please add individual before compound

Ans: As per the reviewers’ suggestion the individual compound was inserted in the content.

Round 2

Reviewer 1 Report

The manuscript can be accepted to publish.

Author Response

Dear reviewer,

thank your for your positive comment and contribution to the high quality of our manuscript.

Reviewer 3 Report

I can not accept the paper in the present form. My specific comments and suggestions for improving the paper are:

Line 289, please define the abbreviation RE

Regarding table 2, Statistical analysis output (ANOVA)  is still missing. Please, make statistical analysis output (ANOVA) for each phytochemicals. Please, in each row insert  letters indicating statistical differences for the different solvent.

Table 3: please, in each row insert  letters indicating statistical differences for the different solvent.

Table 4: idem

Table 5: idem

Table 6: idem

Figure 1:Replace commas with points for decimal.

Figure 2:idem

Figure 3:idem

About my question: Line 301, Why do you use rutin instead of trolox?

Ans: We have performed with plumbagin and Trolox only. In the future we will employ the rutin in the place of Trolox.

Your answer is very confusing…. In the materials and methods remain rutin……please, correct exactly as you have performed the experiment.

Author Response

I can not accept the paper in the present form. My specific comments and suggestions for improving the paper are:

Line 289, please define the abbreviation RE

Ans: We have included the abbreviation for RE – Rutin Equivalent

Regarding table 2, Statistical analysis output (ANOVA) is still missing. Please, make statistical analysis output (ANOVA) for each phytochemicals. Please, in each row insert  letters indicating statistical differences for the different solvent.

Ans: As per the reviewer instruction the Anova is performed and subsets of albha is included in the table.

Table 3: please, in each row insert  letters indicating statistical differences for the different solvent.

Ans: As per the reviewer instruction the Anova is performed and subsets of albha is included in the table.

Table 4: idem

Ans: As per the reviewer instruction the Anova is performed and subsets of albha is included in the table.

Table 5: idem

Ans: As per the reviewer instruction the Anova is performed and subsets of albha is included in the table.

Table 6: idem

Ans: As per the reviewer instruction the Anova is performed and subsets of albha is included in the table.

Figure 1:Replace commas with points for decimal.

Ans: As per the reviewer instruction the Anova is performed and subsets of albha is included in the table. The commas were removed with points for the decimal.

Figure 2:idem

Ans: As per the reviewer instruction the Anova is performed and subsets of albha is included in the table. The commas were removed with points for the decimal.

Figure 3:idem

Ans: As per the reviewer instruction the Anova is performed and subsets of albha is included in the table. The commas were removed with points for the decimal.

About my question: Line 301, Why do you use rutin instead of trolox?

Ans: We have performed with plumbagin and Trolox only. In the future we will employ the rutin in the place of Trolox.

Your answer is very confusing…. In the materials and methods remain rutin……please, correct exactly as you have performed the experiment.

Ans: The Rutin is removed and plumbagin is inserted.